# Tropical Grasses—Annual Crop Intercropping and Adequate Nitrogen Supply Increases Soil Microbial Carbon and Nitrogen

**Karina Batista** [1],* and **Laíze Aparecida Ferreira Vilela** [2]

[1] Instituto de Zootecnia—IZ, Agência Paulista de Tecnologia dos Agronegócios—APTA, 56 Heitor Penteado, St. Centro, Nova Odessa 13380-011, São Paulo, Brazil

[2] Centro de Ciências da Natureza, Universidade Federal de São Carlos, Rodovia Lauri Simões de Barros, km 12—SP-189, Bairro Aracaçú, Buri 18290-000, São Pauo, Brazil; laizevilela@gmail.com

* Correspondence: batistakarin@gmail.com

**Abstract:** In Brazil, grain crops in no-till soybean–maize succession have reduced biodiversity and carbon input into soil. Intercropping is a promising approach to address these problems. This study aimed to evaluate the microbiological quality of soil in conventional and intercropping systems in soybean–maize succession, depending on tropical grass and nitrogen fertilizer uses. The treatments were arranged in a randomized complete block design and a split plot scheme, with four replications. The main plots consisted of the following cropping systems: soybean monoculture–maize monoculture; soybean intercropped with Aruana Guinea grass (*Megathyrsus maximus* cv. Aruana)–maize intercropped with Aruana Guinea grass; and soybean intercropped with Congo grass (*Urochloa ruziziensis* cv. Comun)–maize intercropped with Congo grass. The subplots consisted of nitrogen rates (0, 50, 100, and 150 kg ha$^{-1}$) applied as side-dressing in rows of maize and tropical grass in the autumn–winter season. Our results showed that maize or soybean intercropped with tropical grasses and adequate nitrogen rates favored the entry of microbial carbon and nitrogen, stimulated enzymatic activity, and reduced C-CO$_2$ loss. However, the excess nitrogen supply can nullify the benefits of the intercropping systems. We concluded that the intercropping systems can improve soil microbiological quality in a short time with adequate nitrogen supply.

**Keywords:** Aruana Guinea grass; Congo grass; ecological services; intercropping system; soybean–maize succession; sustainable agriculture

## 1. Introduction

In 2018, the countries composing BRICS, of which Brazil is a part, were responsible for more than 50% of the world's agricultural production [1]. Brazilian grain production has largely used succession between soybeans (*Glycine max*) (summer) and maize (*Zea mays*) (autumn–winter) with no tilling since the 1990s. However, the lack of diversification of this system has reduced biodiversity, carbon input into the soil, and crop yields [2]. One way of addressing these concerns while maintaining the row crop output of these systems is through the establishment and maintenance of another plant between the rows of maize or soybean, a practice known as intercropping.

Intercropping is an agricultural practice of cultivating two or more crops in the same space at the same time, which aims to efficiently match crop demands to the available growth resources and labor. This system improves soil fertility through biological nitrogen fixation with the use of legumes, increases soil conservation through greater ground cover than a monoculture system, and provides better lodging resistance for crops susceptible to lodging than when crops are grown in monoculture [3]. Thus, intercropping systems may enhance soil exploitation by crops and favor labile organic carbon accumulation and, consequently, nutrient cycling [4–7]. However, the selection of an appropriate intercropping system for each case is quite complex as the success of intercropping systems depends on

the interactions between the component species, the available management practices, and the environmental conditions [3].

In recent years, intercropping systems between maize and tropical grasses with no tillage in the autumn–winter season have shown high grain yields and biomass productions in Brazil [8]. Among the tropical grass genera used in these intercropping systems, *Urochloa* and *Megathyrsus* showed good soil coverage. However, tropical grass species to be intercropped with soybeans must be identified. Research [7] showed that soil basal respiration, soil microbial biomass carbon, and soil metabolic quotient are sensitive markers of the effects of soybean–tropical grass intercropping on the soil microbiological activity.

In terms of nutrient cycling, both the amount of plant residue on the soil surface and the availability and lability of nutrients such as carbon and nitrogen in the soil are strongly influenced by nitrogen fertilization [9]. In addition to this, the high demand for nitrogen from maize and tropical grasses for development and productivity highlights the importance of adequately supplying nitrogen in intercropping systems [8]. However, currently, nitrogen fertilization is only performed in planting rows in maize–tropical grass intercropping systems, either in planting or side-dressing, as recommendations consider grain farming [4,10]. Therefore, improving nitrogen fertilization recommendations in maize–tropical grass intercropping, aiming not only for productive gains, but also for soil microbiological quality maintenance, is essential for the sustainability of food systems.

Based on the hypothesis that tropical grasses intercropped with soybeans or maize, combined with efficient nitrogen fertilization, improves plant–soil–microorganism interactions and changes the soil microbiological quality, we evaluated the attributes microbial biomass carbon (MBC), soil basal respiration (SBR), metabolic quotient (qCO$_2$), microbial quotient (qMic), microbial biomass nitrogen (MBN), urease activity (UA), microbial nitrogen to total nitrogen ratio (Nmic:Ntotal), and microbial carbon to microbial nitrogen ratio (Cmic:Nmic) in soybean–maize succession depending on the tropical grass species used and nitrogen fertilization practice.

## 2. Materials and Methods

### 2.1. Characterization and Management of the Experimental Site

The experiment was carried out in southeastern Brazil (22°42′ S, 47°18′ W, and 570 m altitude) in a in a Red-Yellow Argisol–Ultisol [11,12] in an area of degraded grassland composed of Signal grass (*Urochloa decumbens*), Palisade grass (*Urochloa brizantha*), and the Leucaena legume (*Leucaena leucocephala*). According to the classification of Köppen, the local climate is "Aw" type, which stands for rainy tropical forest with rains in the summer and droughts in the winter [13]. From August 2019 to September 2021, soybean was grown without nitrogen application in summer, and maize fertilized with nitrogen as side-dressing was grown in autumn–winter. The crops were named as the first soybean crop (2019/2020) and second soybean crop (2020/2021), as well as the first maize crop (2020) and second maize crop (2021). Figure 1 shows the maximum and minimum temperatures and rainfall during the experimental period.

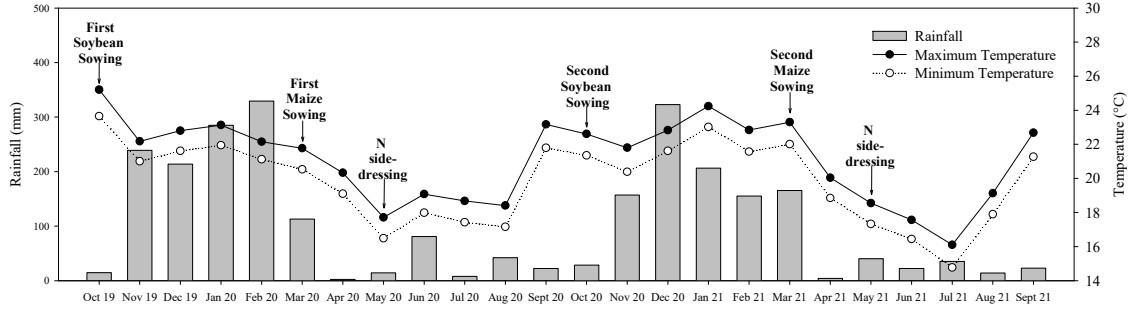

**Figure 1.** Maximum and minimum temperatures and rainfall during the first (October 2019 to September 2020) and second (October 2020 to September 2021) cropping cycle.

Before the beginning of the experiment, soil was sampled to 20 cm depth and analyzed according to the methods described by [14]. The main characteristics were pH ($CaCl_2$) = 4.7; organic matter (colorimetric method) = 30 g $dm^{-3}$; phosphorus (resin) = 4 mg $dm^{-3}$; potassium (resin) = 1.5 $mmol_c$ $dm^{-3}$; calcium (resin) = 10.0 $mmol_c$ $dm^{-3}$; magnesium (resin) = 7.0 $mmol_c$ $dm^{-3}$; potential acidity (H + Al, SMP buffer solution method) = 47 $mmol_c$ $dm^{-3}$; sulfate ($SO_4^{-2}$, turbidimetric method) = 9.0 mg $dm^{-3}$; sum of extractable bases = 19.0 $mmol_c$ $dm^{-3}$; cation exchange capacity = 66.00 $mmol_c$ $dm^{-3}$; and base saturation = 28%, clay = 239 g $kg^{-1}$, silt = 91 g $kg^{-1}$, total sand = 670 g $kg^{-1}$, coarse sand = 120 g $kg^{-1}$, and fine sand = 550 g $kg^{-1}$.

Plowing, harrowing, and liming were carried out before the beginning of the experiment and after sampling of soil. Liming was performed with 2 t $ha^{-1}$ of dolomitic limestone (>12% MgO) and phosphating with 72 kg $ha^{-1}$ of $P_2O_5$ as recommended by [15]. The limestone was distributed in rows and incorporated with a disk plow. The $P_2O_5$ was applied in rows 30 days after liming and incorporated with a disk plow.

## 2.2. Treatments and Experimental Design

The treatments were arranged in a randomized complete block design in a split plot scheme, with four replications. The main plots consisted of the following treatments: (1) soybean monoculture–maize monoculture; (2) soybean intercropped with Aruana Guinea grass (*Megathyrsus maximum* cv. Aruana)–maize intercropped with Aruana Guinea grass; and (3) soybean intercropped with Congo grass (*Urochloa ruziziensis* cv. Comum)–maize intercropped with Congo grass. The subplots consisted of four nitrogen rates (0, 50, 100, and 150 kg $ha^{-1}$) applied as side-dressing in rows of maize and tropical grass in the autumn–winter season about 30 days after planting, when the maize plants had 5–6 fully expanded leaves (Figure 2). The experimental plots were 72 $m^2$ (3.6 m × 20 m).

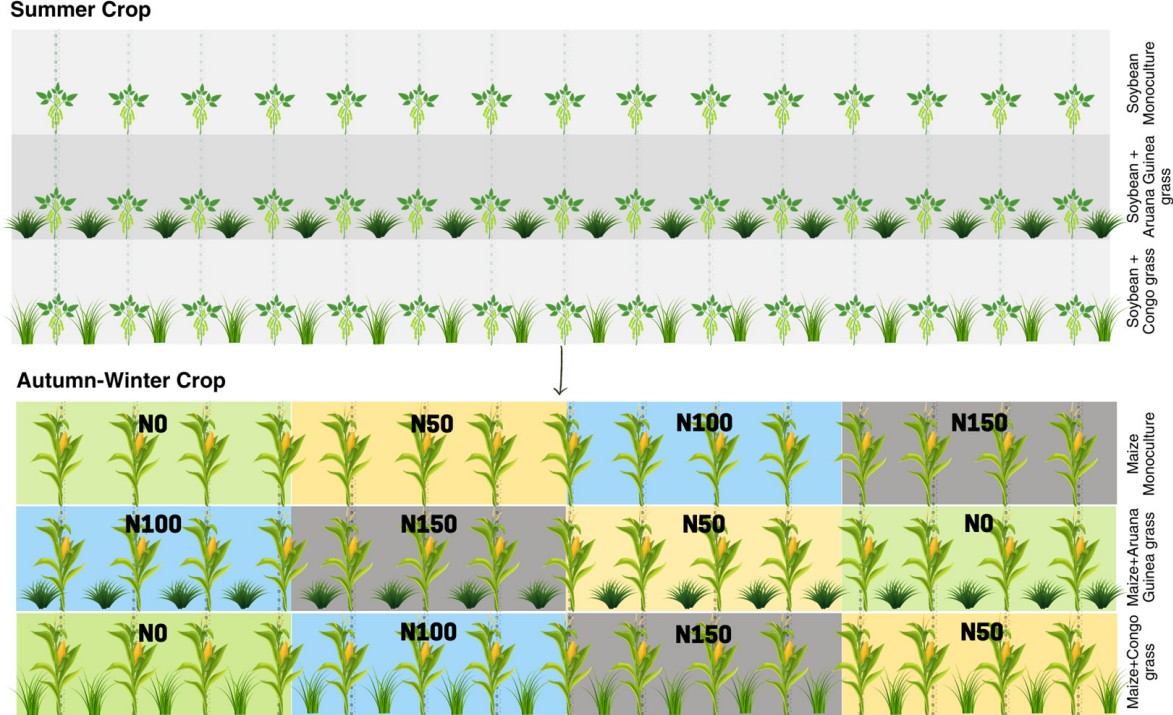

**Figure 2.** Schematic representation of the experimental block design with plots (cropping systems) and subplots (N rates) during the summer (soybean) and autumn–winter (maize) seasons. N0: absence of nitrogen supply, N50: 50 kg $ha^{-1}$, N100: 100 kg $ha^{-1}$, and N150: 150 kg $ha^{-1}$.

Soybean cultivar M6410IPRO (INTACTA RR2 PRO®, Agro Bayer Brazil) was sown in October of each year. Seeds were inoculated with *Bradyrhizobium japonicum*. In intercropping systems soybean and the associated grass were sown simultaneously, using a planter equipped with dispenser boxes separated for large and small seeds. At planting, only

soybean rows were fertilized with 17 kg ha$^{-1}$ of N, 59 kg ha$^{-1}$ of $P_2O_5$, and 34 kg ha$^{-1}$ of $K_2O$ [15]. Soybean rows were spaced 0.45 m apart, with a density of 300,000 plants per hectare. In intercropping systems, soybean and grass rows were spaced 0.225 m apart. Grass seeds had 60% cultural value and were sown at a density of about 6 kg ha$^{-1}$. In all soybean cropping systems, plants (soybean or soybean plus grass) were cut and ensiled at the beginning of soybean maturity, when pods had mature color on the main stem (R7 stage) with a forage harvester (Casale 180 harvester, São Carlos, São Paulo, Brazil).

Maize cultivar AG8061PRO2 (VTPRO2, Agro Bayer Brazil) was sown in March of each year. Maize and grasses were also sown simultaneously, with a planter equipped with dispenser boxes separated for large and small seeds. Maize rows were spaced 0.90 m apart, with a density of 52 thousand plants per hectare. In maize–grass intercropping treatments, maize and grass rows were spaced 0.45 m apart. At planting, only maize rows were fertilized with 30 kg ha$^{-1}$ of N, 50 kg ha$^{-1}$ of $P_2O_5$, and 40 kg ha$^{-1}$ of $K_2O$ [15]. Maize grains were harvested at physiological maturity. Grasses were desiccated with glyphosate herbicide (1.440 g e.a. ha$^{-1}$) in autumn–winter, 30 days before the planting of the next soybean crop. Grass residue was also mowed with a brush cutter (KD170, Araras, São Paulo, Brazil).

### 2.3. Soil Sampling and Analysis

Soil samples were collected in the useful area, at the 0–20 cm depth and between the two central rows of each plot, when the soybean or maize were at the flowering stage with Dutch auger (Sondaterra$^®$ TP—3", Piracicaba, São Paulo, Brazil). After collection and homogenization, the samples were maintained at a temperature of 4 °C until soil microbiological evaluation. The soil microbiological attributes analyzed comprised: (a) microbial biomass carbon (MBC), (b) soil basal respiration (SBR), (c) metabolic quotient (qCO$_2$), (d) microbial quotient (qMic), (e) microbial biomass nitrogen (MBN), (f) urease activity (UA), g) microbial nitrogen to total nitrogen ratio (Nmic:Ntotal), and (h) microbial carbon to microbial nitrogen ratio (Cmic:Nmic).

MBC was extracted from soil samples through the fumigation–extraction method, using chloroform and $K_2SO_4$, respectively. MBC contents were measured through potassium dichromate oxidation and colorimetric titration with 33.3 mM ferrous ammonium sulfate [16]. SBR was obtained by quantifying carbon released in gaseous form ($CO_2$) in soil samples incubated with 0.05 M NaOH and titrating with 0.05 M HCl [17]. qCO$_2$ was determined as the ratio between the C-$CO_2$ content in SBR and MBC content [18]. qMic was calculated as the ratio between the MBC content and total organic carbon (TOC). MBN was determined using the same extract obtained for MBC and quantified through the Kjeldahl method [19]. Lastly, UA was quantified by determining the ammonia released after the incubation of soil samples at 37 °C for 2 h, using a buffer solution with pH:9 [20].

The total amount of N used in the Nmic:N total ratio was determined at maize flowering in both years (autumn–winter). It was extracted through organic nitrogen oxidation with sulfuric acid at high temperatures and quantified through the semi-micro Kjeldahl method [14]. TOC was evaluated at the beginning and end of the experiment. It was extracted using a mixture of sulfuric acid and potassium dichromate and determined by direct reading of green color intensity within the visible spectrum, using a spectrophotometer [14].

### 2.4. Statistical Analysis

Data from the first soybean crop were subjected to analysis of variance (ANOVA) using the SAS software [21], and the means were compared using Tukey's test at 5% probability. Data from the second soybean crop, as well as those of the first and second maize crops, were subjected to analysis of variance through the SASTM GLM procedure at a 5% significance level. The main effects and interactions were studied. Significant interactions were broken down according to the factors involved. The means of each cropping system within each nitrogen rate were compared using Tukey's test, and the effect of nitrogen rates within each cropping system was verified by using regression analysis.

For the main effects in isolation, means were compared by using Tukey's test, and for the effects of nitrogen rates, regression analyses were performed.

## 3. Results

### 3.1. Microbial Biomass Carbon (MBC)

Soybean–Aruana Guinea grass intercropping was the system that most benefited soil MBC incorporation in the first soybean crop (Table 1). Soybean–Aruana Guinea grass intercropping at the nitrogen rate of 50 kg ha$^{-1}$ had the lowest MBC (37.44 µg C g$^{-1}$ soil) in the second soybean crop (Table 2). Maize–Aruana Guinea grass intercropping in the absence of nitrogen supply showed the highest MBC content in the first maize crop (Table 3). MBC in maize–Congo grass intercropping at the nitrogen rate of 50 kg ha$^{-1}$ was equal to that in maize monoculture and reached a content 2.4 times higher than that observed in maize–Aruana Guinea grass intercropping in the first maize crop. Maize–Congo grass intercropping was the system that most favored MBC content up to the nitrogen rate of 100 kg ha$^{-1}$ in the second maize crop (Table 4). However, at the highest nitrogen rate (150 kg ha$^{-1}$), maize–Aruana Guinea grass intercropping was the most efficient system, just as maize monoculture in the second maize crop (Table 4). Furthermore, maize–Aruana Guinea grass intercropping reached the highest MBC (156.61 µg C g$^{-1}$) at the nitrogen rate of 90.40 kg ha$^{-1}$ in the second maize crop, while for maize–Congo grass intercropping, the optimal nitrogen rate was 47.09 kg ha$^{-1}$ (249.67 µg C g$^{-1}$) in the second maize crop (Figure 3a).

**Table 1.** Microbial biomass carbon content (MBC), soil basal respiration (SBR), metabolic quotient (qCO$_2$), microbial quotient (qMic), microbial biomass nitrogen content (MBN), urease activity (UA), and microbial carbon:microbial nitrogen ratio (Cmic:Nmic) of soil at the flowering of the first soybean crop.

| Cropping | MBC | SBR | qCO$_2$ | *q*Mic | MBN | UA | C$_{mic}$:N$_{mic}$ |
|---|---|---|---|---|---|---|---|
| | µg C g$^{-1}$ soil | mg C-CO$_2$ kg$^{-1}$ soil h$^{-1}$ | g C-CO$_2$ g$^{-1}$ C-MBC h$^{-1}$ | | µg N g$^{-1}$ soil | mg N-NH$_4$ g$^{-1}$ soil h$^{-1}$ | |
| SM | 127.76 b | 10.24 a | 0.085 b | 0.010 a | 44.32 a | 4.66 b | 2.90 a |
| S + AGG | 197.34 a | 12.14 a | 0.058 b | 0.010 a | 53.58 a | 7.33 a | 3.68 a |
| S + CG | 74.80 c | 9.97 a | 0.135 a | 0.003 b | 21.13 b | 6.4 a | 3.78 a |
| Means | 133.29 | 10.78 | 0.094 | 0.008 | 39.67 | 6.13 | 3.46 |
| CV | 7.18 [1] | 9.83 [1] | 10.63 [1] | 7.18 [1] | 7.99 [1] | 6.16 [1] | 13.36 [1] |

Means followed by different lowercase letters in the columns differ from each other according to the Tukey test ($p \leq 0.05$). SM = soybean monoculture, S + AGG = soybean–Aruana Guinea grass intercropping, S + CG = soybean–Congo grass intercropping. [1] Coefficient of variation referring to transformed data for log1 + X.

### 3.2. Metabolic Quotient (qCO$_2$)

Soybean–Aruana Guinea grass intercropping reduced qCO$_2$ (32%), benefiting MBC incorporation, while soybean–Congo grass intercropping increased qCO$_2$ (59%), reaching the highest value among all systems in the first soybean crop (Table 1). In the second soybean crop, the nitrogen rate of 50 kg ha$^{-1}$ increased qCO$_2$ in soybean–Aruana Guinea grass intercropping and, hence, reduced the MBC content therein (Table 2). In addition, soybean–Congo grass intercropping and soybean monoculture showed the lowest qCO$_2$ at the nitrogen rate of 100 kg ha$^{-1}$. However, such disturbance did not compromise microbial carbon accumulation in these cropping systems (Table 2). Figure 3d revealed that in soybean–Congo grass intercropping, increasing nitrogen rates significantly reduced qCO$_2$ up to the application rate of 75 kg ha$^{-1}$. The expressive reduction in qCO$_2$ in soybean–Congo grass intercropping was more evident with the application of 75 kg ha$^{-1}$ of nitrogen, which reduced qCO$_2$ by 46%, compared to the absence of nitrogen supply (Figure 3d). Maize–Aruana Guinea grass intercropping at the nitrogen rate of 50 kg ha$^{-1}$ showed higher qCO$_2$ (2.3 times) in the first maize crop (Table 3). However, both the maize–Aruana Guinea

grass and maize–Congo grass intercropping systems responded to nitrogen rates applied as side-dressing in the second maize crop (Table 4 and Figure 3e). Maize–Aruana Guinea grass intercropping showed the lowest soil $qCO_2$ at the nitrogen rate of 75 kg ha$^{-1}$ (0.023 g C-$CO_2$ g$^{-1}$ C-MBC h$^{-1}$), while in maize–Congo grass intercropping, the lowest $qCO_2$ was observed at the nitrogen rate of 40 kg ha$^{-1}$ (0.011 g C-$CO_2$ g$^{-1}$ C-MBC h$^{-1}$) and, above that rate, $qCO_2$ increased, reaching twice the value at the nitrogen rate of 150 kg ha$^{-1}$.

**Table 2.** Microbial biomass carbon content, soil basal respiration, metabolic quotient, microbial quotient, microbial biomass nitrogen content, urease activity, and microbial carbon:microbial nitrogen ratio of soil at the flowering of the second soybean crop.

| Cropping Systems | N Rates (kg ha$^{-1}$) | | | | Means | F Test for Regression | |
| | 0 | 50 | 100 | 150 | | Linear | Quadratic |
| --- | --- | --- | --- | --- | --- | --- | --- |
| | Microbial biomass carbon content (MBC, µg C g$^{-1}$ soil) | | | | | | |
| SM | 236.65 a | 164.18 a | 177.76 a | 131.79 a | 177.60 a | ns | ns |
| S + AGG | 155.29 a | 37.44 b | 326.71 a | 256.39 a | 193.96 a | ns | ns |
| S + CG | 211.23 a | 197.65 a | 297.18 a | 239.89 a | 236.49 a | ns | ns |
| Means | 201.06 | 133.09 | 267.22 | 209.36 | | ns | ns |
| CV | 9.15 [1] | | | | | | |
| | Soil basal respiration (SBR, mg C-$CO_2$ kg$^{-1}$ soil h$^{-1}$) | | | | | | |
| SM | 5.07 b | 3.52 a | 4.91 a | 4.48 a | 4.49 b | ns | ns |
| S + AGG | 5.39 b | 5.48 a | 5.48 a | 4.07 a | 5.05 ab | ns | ns |
| S + CG | 7.78 a | 5.47 a | 4.68 a | 5.43 a | 5.84 a | 0.006 | 0.0010 |
| Means | 6.08 | 4.82 | 5.02 | 4.66 | | ns | ns |
| CV | 8.27 [2] | | | | | | |
| | Metabolic quotient ($qCO_2$, g C-$CO_2$ g$^{-1}$ C-MBC h$^{-1}$) | | | | | | |
| SM | 0.02 a | 0.03 b | 0.03 b | 0.09 a | 0.04 a | ns | ns |
| S + AGG | 0.04 a | 0.15 a | 0.15 a | 0.02 a | 0.06 a | ns | ns |
| S + CG | 0.04 a | 0.03 b | 0.02 b | 0.02 a | 0.03 a | 0.0022 | 0.0003 |
| Means | 0.03 | 0.07 | 0.07 | 0.04 | | ns | ns |
| CV | 32.38 [2] | | | | | | |
| | Microbial biomass nitrogen content (MBN, µg N g$^{-1}$ soil) | | | | | | |
| SM | 31.85 a | 119.53 a | 67.89 a | 62.46 a | 70.43 a | ns | ns |
| S + AGG | 32.38 a | 76.66 a | 76.66 a | 78.74 a | 53.38 a | ns | ns |
| S + CG | 30.27 a | 51.09 a | 68.61 a | 87.32 a | 59.32 a | 0.0004 | ns |
| Means | 31.50 | 82.42 | 71.05 | 76.17 | | ns | ns |
| CV | 10.7 [1] | | | | | | |
| | Urease activity (UA, mg N-$NH_4$ g$^{-1}$ soil h$^{-1}$) | | | | | | |
| SM | 2.6 a | 5.04 a | 4.17 a | 3.01 b | 3.71 a | ns | 0.0378 |
| S + AGG | 1.40 a | 2.17 a | 2.17 b | 3.43 b | 2.30 b | 0.0398 | ns |
| S + CG | 2.87 a | 2.14 a | 3.24 ab | 5.37 a | 3.40 ab | 0.0031 | 0.0002 |
| Means | 2.30 | 3.12 | 3.19 | 3.94 | | ns | ns |
| CV | 19.07 [2] | | | | | | |
| | Microbial carbon:microbial nitrogen ratio (Cmic:Nmic) | | | | | | |
| SM | 8.53 a | 1.30 b | 5.37 b | 2.94 a | 4.53 a | ns | ns |
| S + AGG | 6.99 a | 0.66 b | 15.94 a | 3.60 a | 6.80 a | ns | ns |
| S + CG | 9.62 a | 4.99 a | 4.38 b | 2.83 a | 5.46 a | 0.0225 | ns |
| Means | 8.38 | 2.32 | 8.56 | 3.12 | | ns | ns |
| CV | 30.76 [1] | | | | | | |

Means followed by different lowercase letters in the columns differ from each other according to the Tukey test ($p \leq 0.05$). SM = soybean monoculture, S + AGG = soybean–Aruana Guinea grass intercropping, S + CG = soybean–Congo grass intercropping. ns: not significant ($p > 0.05$). [1] Coefficient of variation referring to transformed data for log1 + X. [2] Coefficient of variation referring to transformed data for $\sqrt{X}$.

**Table 3.** Microbial biomass carbon content, soil basal respiration, metabolic quotient, microbial quotient, microbial biomass nitrogen content (MBN), urease activity, microbial biomass nitrogen:total soil nitrogen ratio, and microbial carbon:microbial nitrogen ratio of soil at the flowering of the first maize crop.

| Cropping Systems | N Rates (kg ha$^{-1}$) | | | | Means | F Test for Regression | |
| --- | --- | --- | --- | --- | --- | --- | --- |
| | 0 | 50 | 100 | 150 | | Linear | Quadratic |
| Microbial biomass carbon content (MBC, µg C g$^{-1}$ soil) | | | | | | | |
| MM | 157.7 b | 121.42 a | 117.12 a | 100.93 a | 124.31 a | ns | ns |
| M + AGG | 281.2 a | 59.53 b | 175.02 a | 103.22 a | 154.75 a | ns | ns |
| M + CG | 149.26 b | 140.66 a | 150.86 a | 197.82 a | 159.65 a | ns | ns |
| Means | 196.08 | 107.21 | 147.67 | 133.99 | | ns | ns |
| CV | 8.45 [1] | | | | | | |
| Soil basal respiration (SBR, mg C-CO$_2$ kg$^{-1}$ soil h$^{-1}$) | | | | | | | |
| MM | 9.34 a | 9.23 a | 9.26 a | 9.65 a | 9.37 a | ns | 0.0117 |
| M + AGG | 9.75 a | 9.17 a | 9.20 a | 9.45 a | 9.39 a | ns | ns |
| M + CG | 9.23 a | 9.04 a | 9.1 a | 8.88 a | 9.07 a | ns | ns |
| Means | 9.44 | 9.15 | 9.19 | 9.33 | | ns | ns |
| CV | 2.54 [1] | | | | | | |
| Metabolic quotient (qCO$_2$, g C-CO$_2$ g$^{-1}$ C-MBC h$^{-1}$) | | | | | | | |
| MM | 0.07 a | 0.08 b | 0.08 a | 0.22 a | 0.11 a | ns | ns |
| M + AGG | 0.03 a | 0.17 a | 0.06 a | 0.09 a | 0.09 a | ns | ns |
| M + CG | 0.06 a | 0.07 b | 0.06 a | 0.05 a | 0.06 a | ns | ns |
| Means | 0.06 | 0.10 | 0.07 | 0.12 | | ns | ns |
| CV | 26.92 [2] | | | | | | |
| Microbial biomass nitrogen content (MBN, µg N g$^{-1}$ soil) | | | | | | | |
| MM | 47.90 a | 28.09 b | 101.61 a | 109.88 a | 71.87 a | 0.0091 | ns |
| M + AGG | 23.60 a | 73.46 a | 12.20 b | 33.64 b | 35.72 b | ns | ns |
| M + CG | 25.22 a | 32.32 b | 26.82 b | 23.61 b | 26.99 b | ns | ns |
| Means | 32.24 | 44.62 | 46.88 | 55.71 | | ns | ns |
| CV | 13.76 [1] | | | | | | |
| Urease activity (UA, mg N-NH$_4$ g$^{-1}$ soil h$^{-1}$) | | | | | | | |
| MM | 3.76 ab | 2.64 a | 7.33 a | 3.83 a | 4.39 a | ns | ns |
| M + AGG | 6.07 a | 2.59 a | 1.6 b | 1.5 b | 2.95 ab | 0.0001 | 0.0025 |
| M + CG | 2.63 b | 2.77 a | 2.63 b | 2.87 ab | 2.72 b | ns | ns |
| Means | 4.15 | 2.67 | 3.86 | 2.74 | | ns | ns |
| CV | 16.56 [2] | | | | | | |
| Microbial biomass nitrogen:total soil nitrogen ratio (Nmic:Ntotal) | | | | | | | |
| MM | 0.03 a | 0.02 b | 0.07 a | 0.07 a | 0.05 a | 0.0035 | 0.0164 |
| M + AGG | 0.02 a | 0.05 a | 0.01 b | 0.03 ab | 0.03 b | ns | ns |
| M + CG | 0.02 a | 0.02 b | 0.02 b | 0.02 b | 0.02 b | ns | ns |
| Means | 0.02 | 0.03 | 0.03 | 0.04 | | ns | ns |
| CV | 26.16 [2] | | | | | | |
| Microbial carbon:microbial nitrogen ratio (Cmic:Nmic) | | | | | | | |
| MM | 4.52 a | 5.34 a | 1.27 b | 0.93 b | 3.01 b | 0.0079 | 0.0292 |
| M + AGG | 13.98 a | 0.92 a | 14.19 a | 3.88 ab | 8.24 a | ns | ns |
| M + CG | 8.06 a | 4.43 a | 5.65 b | 10.53 a | 7.17 ab | ns | ns |
| Means | 8.85 | 3.56 | 7.04 | 5.11 | | ns | ns |
| CV | 24.62 [3] | | | | | | |

Means followed by different lowercase letters in the columns differ from each other according to the Tukey test ($p \leq 0.05$). MM = maize monoculture, M + AGC = maize–Aruana Guinea grass intercropping, M + CG = maize–Congo grass intercropping. ns: not significant ($p > 0.05$). [1] Coefficient of variation referring to transformed data for logX. [2] Coefficient of variation referring to transformed data for $\sqrt{X}$. [3] Coefficient of variation referring to transformed data for logX + 1.

**Table 4.** Microbial biomass carbon content, soil basal respiration, metabolic quotient, microbial quotient, microbial biomass nitrogen content, urease activity, microbial biomass nitrogen:total soil nitrogen ratio, and microbial carbon:microbial nitrogen ratio of soil at the flowering of the second maize crop.

| Cropping Systems | N Rates (kg ha$^{-1}$) | | | | Means | F Test for Regression | |
|---|---|---|---|---|---|---|---|
| | 0 | 50 | 100 | 150 | | Linear | Quadratic |
| Microbial biomass carbon content (MBC, µg C g$^{-1}$ soil) | | | | | | | |
| MM | 183.7 ab | 172.93 b | 195.82 ab | 126.18 a | 169.7 ab | ns | ns |
| M + AGG | 91.39 b | 143.11 b | 156.37 b | 128.13 a | 129.75 b | 0.0511 | 0.0001 |
| M + CG | 216.87 a | 243.74 a | 211.18 a | 80.61 b | 188.10 a | 0.0008 | 0.0001 |
| Means | 163.99 | 153.05 | 169.52 | 111.64 | | ns | ns |
| CV | 9.47 [1] | | | | | | |
| Soil basal respiration (SBR, mg C-CO$_2$ kg$^{-1}$ soil h$^{-1}$) | | | | | | | |
| MM | 3.11 a | 3.12 a | 3.08 a | 3.28 a | 3.15 a | ns | ns |
| M + AGG | 3.09 a | 3.02 a | 3.23 a | 3.17 a | 3.13 a | ns | ns |
| M + CG | 3.54 a | 3.02 a | 3.23 a | 3.04 a | 3.42 a | ns | ns |
| Means | 3.25 | 3.05 | 3.18 | 3.17 | | ns | ns |
| CV | 6.43 [2] | | | | | | |
| Metabolic quotient (qCO$_2$, g C-CO$_2$ g$^{-1}$ C-MBC h$^{-1}$) | | | | | | | |
| MM | 0.02 b | 0.02 a | 0.01 b | 0.03 ab | 0.02 a | ns | ns |
| M + AGG | 0.03 a | 0.02 a | 0.02 a | 0.02 b | 0.03 a | 0.0537 | 0.0005 |
| M + CG | 0.02 b | 0.02 a | 0.02 a | 0.04 a | 0.02 a | 0.0040 | 0.0001 |
| Means | 0.02 | 0.02 | 0.02 | 0.03 | | ns | ns |
| CV | 10.17 [2] | | | | | | |
| Microbial quotient (qMic) | | | | | | | |
| MM | 0.011 ab | 0.009 b | 0.013 a | 0.009 a | 0.012 ab | ns | ns |
| M + AGG | 0.009 b | 0.009 b | 0.010 b | 0.009 a | 0.008 b | 0.0242 | 0.0003 |
| M + CG | 0.0141 a | 0.0148 a | 0.0133 a | 0.0051 b | 0.012 a | 0.0008 | 0.0001 |
| Means | 0.010 | 0.011 | 0.012 | 0.007 | | ns | ns |
| CV | 8.60 [2] | | | | | | |
| Microbial biomass nitrogen content (MBN, µg N g$^{-1}$ soil) | | | | | | | |
| MM | 62.47 a | 51.30 a | 53.57 a | 29.90 a | 49.31 a | 0.0001 | 0.0001 |
| M + AGG | 19.73 b | 40.58 b | 27.91 b | 25.85 ab | 28.52 b | ns | 0.0101 |
| M + CG | 59.38 a | 40.58 b | 27.91 b | 19.59 b | 46.60 a | 0.0001 | 0.0001 |
| Means | 47.19 | 44.15 | 36.46 | 25.11 | | ns | ns |
| CV | 5.89 [1] | | | | | | |
| Urease activity (UA, mg N-NH$_4$ g$^{-1}$ soil h$^{-1}$) | | | | | | | |
| MM | 8.58 a | 7.56 a | 6.70 a | 5.83 a | 7.17 a | 0.0057 | 0.0252 |
| M + AGG | 5.30 b | 6.11 a | 4.87 a | 4.94 a | 5.30 b | ns | ns |
| M + CG | 6.79 ab | 6.11 a | 4.87 a | 4.92 a | 5.87 b | ns | ns |
| Means | 6.89 | 6.59 | 5.48 | 5.23 | | ns | ns |
| CV | 11.48 [1] | | | | | | |
| Microbial biomass nitrogen: total soil nitrogen ratio (Nmic:Ntotal) | | | | | | | |
| MM | 0.05 a | 0.03 a | 0.04 a | 0.02 a | 0.04 a | ns | ns |
| M + AGG | 0.01 b | 0.03 a | 0.02 b | 0.02 a | 0.02 b | ns | ns |
| M + CG | 0.04 a | 0.04 a | 0.03 ab | 0.01 a | 0.03 ab | 0.0001 | 0.0001 |
| Means | 0.03 | 0.03 | 0.03 | 0.02 | | ns | ns |
| CV | 13.54 [2] | | | | | | |
| Microbial carbon:microbial nitrogen ratio (Cmic:Nmic) | | | | | | | |
| MM | 2.94 b | 3.34 a | 3.66 b | 4.25 a | 3.55 b | 0.0280 | ns |
| M + AGG | 4.72 a | 3.53 a | 5.60 a | 5.07 a | 4.73 a | ns | ns |
| M + CG | 3.66 ab | 4.18 a | 4.46 b | 4.12 a | 4.11 ab | ns | ns |
| Means | 3.77 | 3.68 | 4.57 | 4.48 | | ns | ns |
| CV | 8.91 [2] | | | | | | |

Means followed by different lowercase letters in the columns differ from each other according to the Tukey test ($p \leq 0.05$). MM = maize monoculture, M + AGC = maize–Aruana Guinea grass intercropping, M + CG = maize–Congo grass intercropping. ns: not significant ($p > 0.05$). [1] Coefficient of variation referring to transformed data for logX. [2] Coefficient of variation referring to transformed data for $\sqrt{X}$.

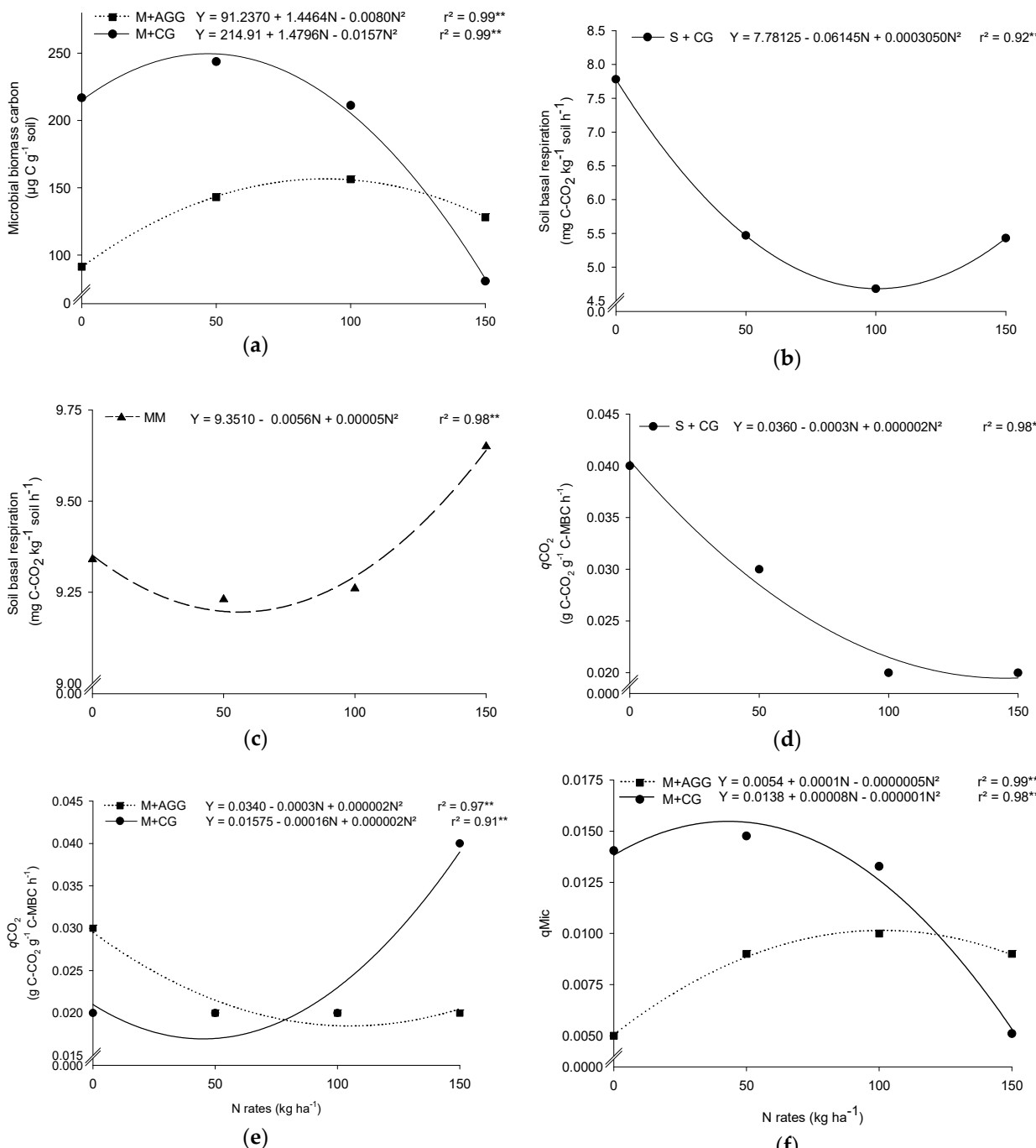

**Figure 3.** (**a**) Microbial biomass carbon content in the second maize crop, (**b**) soil basal respiration in the second soybean crop, (**c**) soil basal respiration in the first maize crop, (**d**) metabolic quotient in the second soybean crop, (**e**) metabolic quotient in the second maize crop, (**f**) microbial quotient in the second maize crop. S + CG: soybean−Congo grass intercropping, MM: maize monoculture system, M + AGG: maize−Aruana Guinea grass intercropping, and M + CG: maize−Congo grass intercropping. * Significant at $p \leq 0.05$. ** Significant at $p \leq 0.01$.

### 3.3. Microbial Quocient (qMic)

Soybean monoculture and soybean–Aruana Guinea grass intercropping showed qMic values higher (70%) than soybean–Congo grass intercropping in the first soybean crop (Table 1). Maize–Congo grass intercropping in the second maize crop, at all nitrogen rates applied as side-dressing (0 to 150 kg ha$^{-1}$), was the system that most increased soil

qMic, not differing from maize monoculture in the absence of nitrogen supply and at the nitrogen rate of 100 kg ha$^{-1}$ (Table 4). Furthermore, at the nitrogen rate of 100 kg ha$^{-1}$, maize–Aruana Guinea grass intercropping promoted the highest soil qMic (0.010), while the highest soil qMic (0.008) in maize–Congo grass intercropping occurred at the nitrogen rate of 44.4 kg ha$^{-1}$ (Figure 3f). Above this nitrogen rate, qMic significantly reduced, reaching 0.003 at the nitrogen rate of 150 kg ha$^{-1}$ (Figure 3f).

### 3.4. Microbial Biomass Nitrogen (MBN)

Soybean monoculture and soybean–Aruana Guinea grass intercropping provided greater MBN in the first soybean crop (Table 1). MBN increased linearly with the nitrogen rates in the soybean–Congo grass intercropping in the second soybean crop (Table 2 and Figure 4a). In the first maize crop (Table 3), MBN increased linearly in maize monoculture as a function of the nitrogen side-dressing rate increasing (Figure 4b). Furthermore, in the absence of nitrogen as side-dressing, cropping systems did not affect MBN. However, at the nitrogen rate of 50 kg ha$^{-1}$, maize–Aruana Guinea grass intercropping more than doubled MBN compared to the other cropping systems. Nevertheless, maize monoculture applied with high nitrogen rates (100 and 150 kg ha$^{-1}$) was more efficient in incorporating nitrogen into the microbial biomass. In the second maize crop (Table 4), in maize monoculture, the highest MBN (60.50 µg N g$^{-1}$ soil) occurred at the nitrogen rate of 1.27 kg ha$^{-1}$ and, above this rate, it reduced (Figure 4c). Maize–Aruana Guinea grass intercropping showed the highest MBN (35.63 µg N g$^{-1}$ soil) at the nitrogen rate of 77.15 kg ha$^{-1}$ (Figure 4c) and, despite a subsequent reduction, results were higher than those observed in the absence of nitrogen. Maize–Congo grass intercropping showed the highest MBN (61.62 µg N g$^{-1}$ soil) at the nitrogen rate of 29.14 kg ha$^{-1}$ (Figure 4c). These results showed that maize–Aruana Guinea grass intercropping was the system that most benefited from nitrogen rates applied as side-dressing. MBN results also revealed that, in the absence of nitrogen, maize–Congo grass intercropping matched maize monoculture, where MBN was higher. At the nitrogen rates of 50 and 100 kg ha$^{-1}$, intercropping systems could not overcome the MBN observed in maize monoculture. However, at the nitrogen rate of 150 kg ha$^{-1}$, the highest MBN was observed in maize monoculture, which did not differ from maize–Aruana Guinea grass intercropping.

### 3.5. Urease Activity (UA)

In terms of enzymatic activity, both intercropping systems favored UA compared to soybean monoculture, which had an enzymatic activity lower (32%) than the other cropping systems in the first soybean crop (Table 1). In the second soybean crop, unlike the first soybean crop, UA was not affected by cropping systems in the absence of nitrogen supply (Table 2). At the highest nitrogen rates (100 and 150 kg ha$^{-1}$), enzymatic activity varied between the cropping systems (Table 2). At the nitrogen rate of 100 kg ha$^{-1}$, soybean monoculture and soybean–Congo grass intercropping showed higher enzymatic activity. At the nitrogen rate of 150 kg ha$^{-1}$, UA was 67% higher in soybean–Congo grass intercropping than in the other cropping systems. Furthermore, regression analysis showed that all cropping systems in the second soybean crop responded to the nitrogen rates applied in the maize crop (Table 2). In soybean monoculture, UA increased up to the nitrogen rate of 68 kg ha$^{-1}$ (Figure 4d). In soybean–Aruana Guinea grass intercropping, increasing nitrogen rates positively and linearly increased enzyme activity (Figure 4d). Finally, in soybean–Congo grass intercropping, the nitrogen rate of 43 kg ha$^{-1}$ promoted the lowest UA, while the nitrogen rate of 150 kg ha$^{-1}$ showed a UA two times higher than in the absence of nitrogen supply (Figure 4d). In the first maize crop, in the absence of nitrogen supply, UA in maize–Aruana Guinea grass intercropping was statistically equal to that in maize monoculture (Table 3). The nitrogen rates of 100 and 150 kg ha$^{-1}$ increased UA in maize monoculture; therefore, this system benefited from a greater nitrogen supply. On the other hand, high nitrogen rates did not favor enzymatic activity in intercropping systems, but the nitrogen rate of 150 kg ha$^{-1}$ in maize–Congo grass intercropping showed a statistically

intermediate value (2.87 mg N-NH4 g$^{-1}$ soil h$^{-1}$). Table 3 shows the significance for the interaction between maize–Aruana Guinea grass intercropping and rates of nitrogen applied as side-dressing (Figure 4e), and also shows that the lowest enzymatic activity occurred at the nitrogen rate of 133.66 kg ha$^{-1}$, wherein enzymatic activity reduced by 90% (0.63 mg N-NH4 g$^{-1}$ soil h$^{-1}$). In the second maize crop, UA showed significance for the interaction between maize monoculture and rates of nitrogen applied as side-dressing (Table 4), with the lowest value being reached at nitrogen rates outside the range studied here (Figure 4f). UA results also showed that enzymatic activity only showed differences between cropping systems in the absence of nitrogen supply, with higher UA in maize monoculture, which did not differ from maize–Congo grass intercropping.

### 3.6. Microbial Carbon to Microbial Nitrogen Ratio (Cmic:Nmic Ratio)

In the second soybean crop, soybean–Congo grass intercropping at the nitrogen rate of 50 kg ha$^{-1}$ had a Cmic:Nmic ratio (4.99) four times higher than in the other cropping systems (Table 2). At the nitrogen rate of 100 kg ha$^{-1}$, soybean–Aruana Guinea grass intercropping revealed a Cmic:Nmic ratio (15.94) higher than the other cropping systems. Conversely, in the absence of nitrogen supply and at the rate of 150 kg ha$^{-1}$, there were no significant differences between the cropping systems. Finally, Figure 5a shows that the Cmic:Nmic ratio decreased linearly with increasing nitrogen rates in the soybean–Congo grass intercropping (Figure 5a). In the first maize crop, at nitrogen rates of 100, maize–Aruana Guinea grass intercropping increased the Cmic:Nmic ratio compared to maize monoculture and maize–Congo grass intercropping (Table 3). Maize monoculture, in turn, responded to the increasing nitrogen rates applied as side-dressing, with the highest Cmic:Nmic ratio being observed at the nitrogen rate of 61 kg ha$^{-1}$ (Figure 5b). In the second maize crop, the Cmic:Nmic ratio showed significance for the interaction between maize monoculture and rates of nitrogen applied as side-dressing (Table 4), and this ratio increased linearly as a function of the nitrogen rates (Figure 5c). Furthermore, in the absence of nitrogen supply, maize–Aruana Guinea grass intercropping showed the highest Cmic:Nmic ratio, not differing from maize–Congo grass intercropping. On the other hand, at the nitrogen rate of 100 kg ha$^{-1}$, maize–Aruana Guinea grass intercropping showed the highest Cmic:Nmic ratio, differing from the other cropping systems.

### 3.7. Microbial Biomass Nitrogen to Total Soil Nitrogen Ratio (Nmic:Ntotal Ratio)

The nitrogen rate of 50 kg ha$^{-1}$ doubled the Nmic:Ntotal ratio in maize–Aruana Guinea grass intercropping in the first maize crop, similarly to what was observed for MBN (Table 3). At the nitrogen rates of 50 kg ha$^{-1}$ and 100 kg ha$^{-1}$, maize monoculture was more efficient in terms of increasing the Nmic:Ntotal ratio in the soil, similar to what was observed for UA and MBN in the first maize crop (Table 3). Furthermore, the Nmic:Ntotal ratio showed significance for the interaction between maize monoculture and rates of nitrogen applied as side-dressing, and Figure 5d reveals that the highest Nmic:Ntotal ratio occurred at a nitrogen rate higher than those studied (500 kg ha$^{-1}$). In the second maize crop (Table 4), maize–Congo grass intercropping showed the highest Nmic:Ntotal ratio (0.039) at the nitrogen rate of 22.5 kg ha$^{-1}$ (Figure 5e). At the highest nitrogen rate (150 kg ha$^{-1}$), maize–Congo grass intercropping showed a reduction (83.8%) in the Nmic:Ntotal ratio regarding the optimal nitrogen rate (22.55 kg ha$^{-1}$). Furthermore, in the absence of nitrogen supply and at the nitrogen rate of 100 kg ha$^{-1}$, the Nmic:Ntotal ratio did not show a significant difference between maize monoculture and maize–Congo grass intercropping (Table 4).

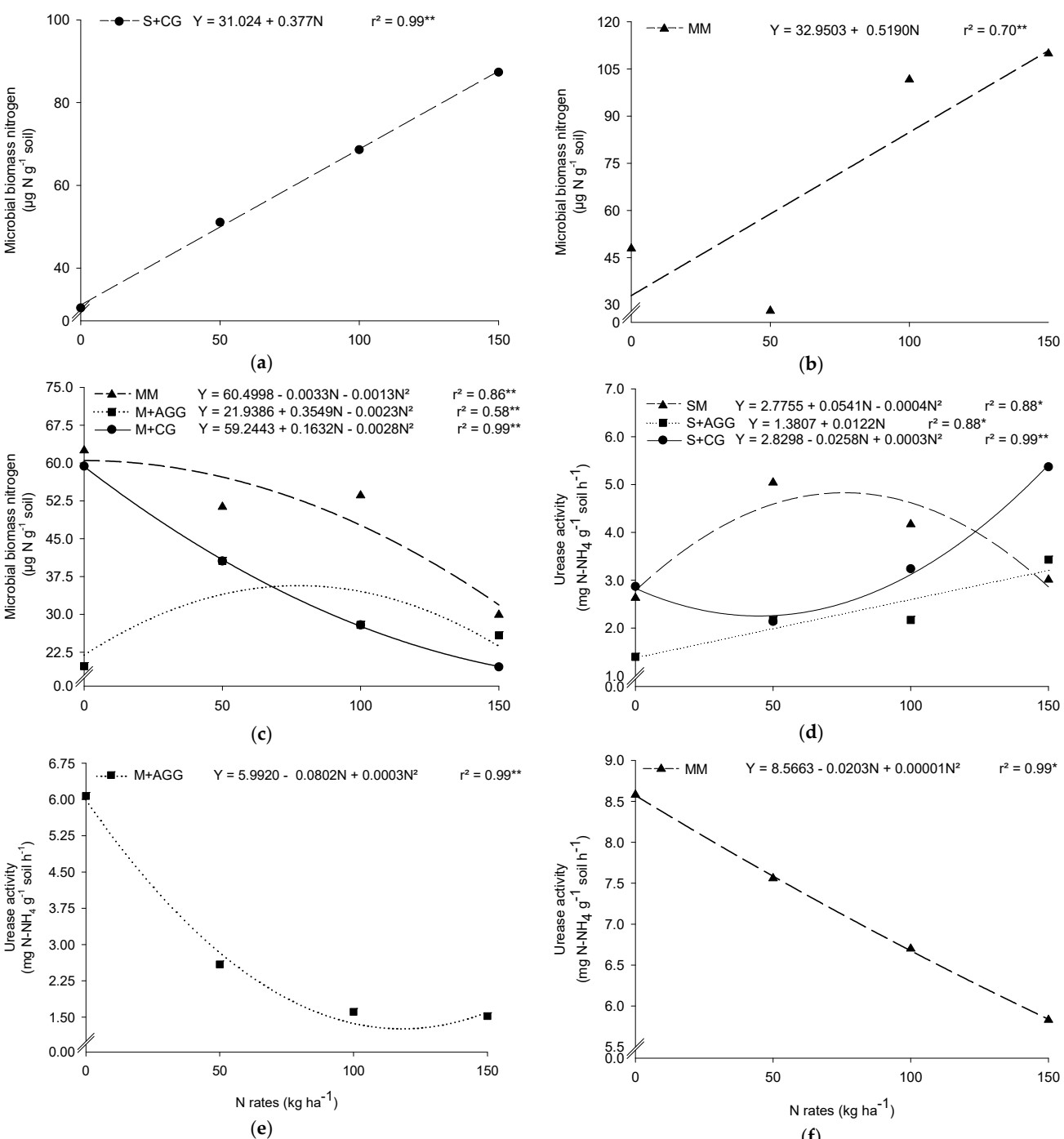

**Figure 4.** (**a**) Microbial biomass nitrogen in the second soybean crop, (**b**) microbial biomass nitrogen in the first maize crop, (**c**) microbial biomass nitrogen in the second maize crop, (**d**) urease activity in the second soybean crop, (**e**) urease activity in the first maize crop, and (**f**) urease activity in the second maize crop as a function of nitrogen side-dressing application rate at the flowering stage. SM: soybean monoculture system, S + AGG: soybean−Aruana Guinea grass intercropping, S + CG: soybean−Congo grass intercropping, MM: maize monoculture system, M + AGG: maize−Aruana Guinea grass intercropping, and M + CG: maize−Congo grass intercropping. * Significant at $p \leq 0.05$. ** Significant at $p \leq 0.01$.

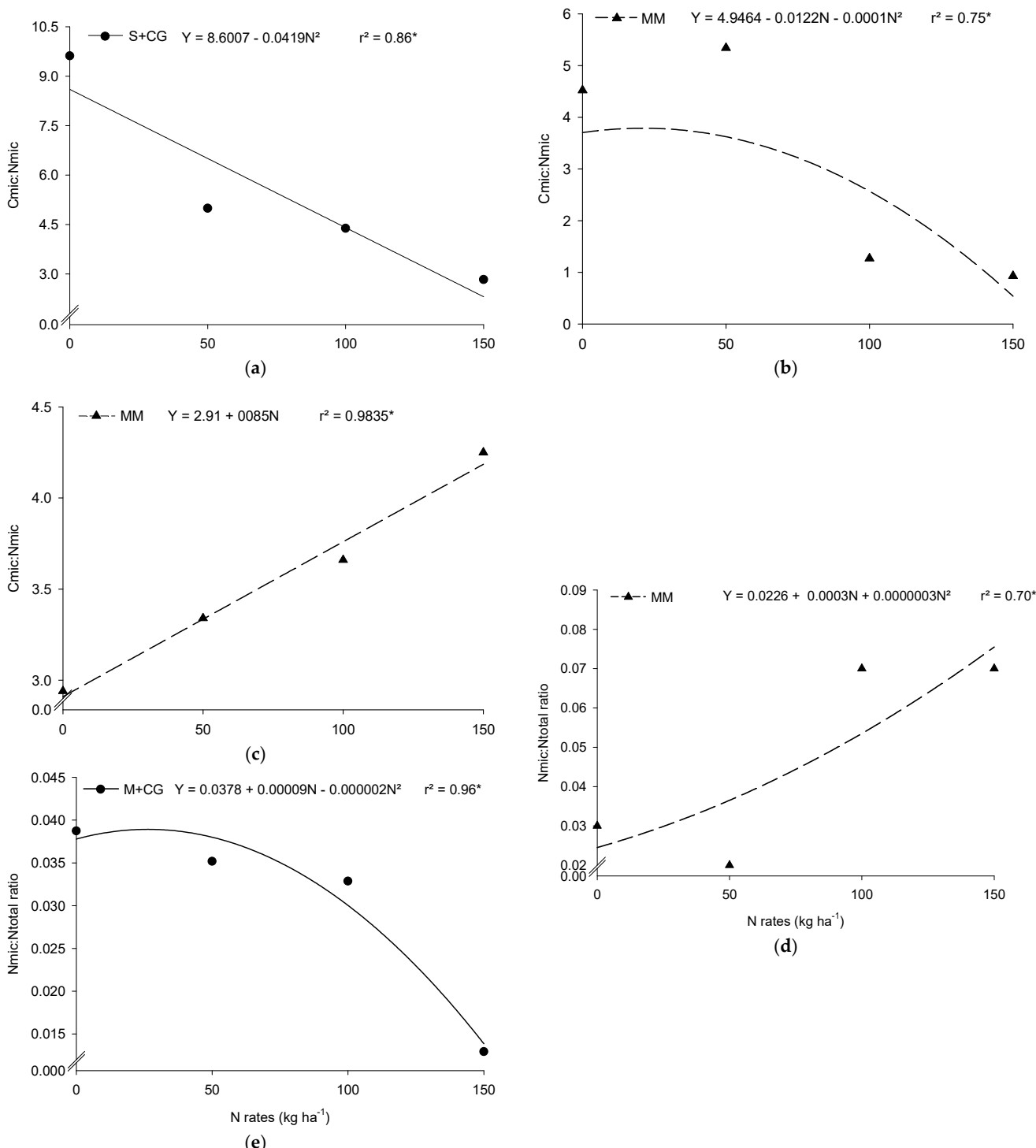

**Figure 5.** (**a**) Microbial carbon to microbial nitrogen ratio in the second soybean crop, (**b**) microbial carbon to microbial nitrogen ratio in the first maize crop, (**c**) microbial carbon to microbial nitrogen ratio in the second maize crop, (**d**) microbial biomass nitrogen:total soil nitrogen ratio in the first maize crop, and (**e**) microbial biomass nitrogen:total soil nitrogen ratio in the second maize crop as function of nitrogen side-dressing application rates at flowering stage. S + CG: soybean − Congo grass intercropping, MM: maize monoculture system, M + CG: maize − Congo grass intercropping. * Significant at $p \leq 0.05$.

## 4. Discussion

Soil carbon and nitrogen stability are essential for microbial balance and nutrient cycling efficiency. In our study, maize or soybean intercropping systems with tropical grasses supplied with suitable nitrogen rates favored the entry of microbial carbon and nitrogen, stimulated enzymatic activity, and reduced C-$CO_2$ loss (Tables 2–4). However, the benefits from nitrogen rates and intercropping systems on increasing microbial carbon and nitrogen varied between soybean and maize crops. Thus, systems combining cover crop and nitrogen use efficiency for grain production can stimulate the microbial community and ensure the sustainability of ecosystem services [22].

Soybean–Aruana Guinea grass intercropping was the system that contributed most to microbial carbon content in the soil, in the first soybean crop, in which there was no nitrogen application (Table 1). Similarly, the soil $qCO_2$ increase in soybean–Congo grass intercropping reduced MBC content and, consequently, reduced qMic (Table 1). Comparing only cropping systems (without nitrogen addition), these findings suggest that C loss was intensified by soybean–Congo grass intercropping without adequate nitrogen supply due to increase in oxidative processes, leading the microbiota to release C into the atmosphere at the expense of its incorporation into microbial biomass. Thus, the use of Congo grass as a cover crop reduced the energy efficiency of microbial communities in the soil [23,24] and the use of Aruana guinea grass carbon savings [25]. It is worth mentioning that in each soybean harvest, the entire plant was harvested for silage. Thus, there was no influence of soybean residues on subsequent maize crops, both in the first and in the second maize crop.

Maize–Aruana Guinea grass intercropping in the absence of nitrogen supply showed higher MBC content than the other cropping systems in the first maize crop (Table 3). However, at the nitrogen rate of 50 kg ha$^{-1}$, this content decreased drastically, evidencing the negative effect of the nitrogen supply in this intercropping, even at lower rates (Table 3). This showed that when N availability increases, plants invest less C into the roots and mycorrhiza because less effort is required to acquire this resource from soil [26]. On the other hand, the nitrogen rate of 50 kg ha$^{-1}$ favored an increase in MBC under maize–Congo grass intercropping, which may have been caused by a reduced $qCO_2$, making it as efficient as maize monoculture in incorporating C into microbial biomass [7,24]. In the second soybean crop, nitrogen supply (50 kg ha$^{-1}$) also negatively affected MBC content in soybean–Aruana Guinea grass intercropping (−79%), in addition to increasing $qCO_2$ (Table 2). Additionally, only the highest nitrogen rate (150 kg ha$^{-1}$) minimized the microbial community stress, a fact evidenced by the comparison of the $qCO_2$ of the soybean–Aruana Guinea grass intercropping with the other cropping systems (Table 2). These data suggest that this intercropping system favors C increases more efficiently in the absence of nitrogen. On the other hand, in soybean–Congo grass intercropping, increasing nitrogen rates promoted a reduction in soil $qCO_2$ (Figure 3d); therefore, nitrogen application improved the soil microbiota balance in the carbon waste decomposition processes, demonstrating the higher N demand for these intercropping systems. According to [27], Congo grass has faster decomposition of its plant residues than *Cajanus cajan* and Sorghum bicolor, and the authors related these results to lower lignin concentrations and lignin:nitrogen ratios in the plant tissues of this grass. Another important point is that low-lignin crop residues are less recalcitrant to microbial degradation and, therefore, have a faster decomposition [5], which may have contributed to the increase in MBC in this study.

There was an inversion of the response from intercropping systems regarding the MBN content in the second maize crop (Table 4). Maize–Aruana Guinea grass intercropping decreased MBN with increasing nitrogen supply. This outcome suggests that maize–Aruana Guinea grass intercropping, due to the stress caused by nitrogen fertilization, requires a lower nitrogen supply to rebalance the MBN content in the soil. The findings of [26] suggest that in this case, maize–Aruana Guinea grass intercropping promoted an increase in plant biodiversity and interspecific interaction, but this was broken with the highest nitrogen supply.

The response of the maize–Aruana Guinea grass intercropping in the last cycle showed that, as the rates were applied to the soil, there was a continuous reduction in MBN (Table 4). This is evident by observing the response of Aruana Guinea grass intercropping in the first soybean crop (Table 1), where no nitrogen rate had been applied, and its behavior after the addition of nitrogen in subsequent crops (starting from the first maize crop) (Tables 2–4). Moreover, the reduction in MBC content in this study was accompanied by an increase in $qCO_2$, which resulted in lower values of qMic. This information reinforces the suggestion that the reduction in MBC content in all crops occurred due to an increase in energy consumption by microbial communities for biological maintenance [2,7,9,23,28]. Higher $qCO_2$ values highlight environments with a less stable microbiota [18]; therefore, the ecophysiological state of the microbial community is unbalanced [24].

The MBN content in the soil during the first soybean crop was impaired by the soybean–Congo grass intercropping, with a 57% reduction in relation to the other cropping systems (Table 1). However, the soybean–Aruana Guinea grass intercropping maintained the NBM content compared to the monoculture system, in addition to increasing the urease activity (+57%), suggesting that the nitrogen supply or immobilization in the soil in this intercropping was closely regulated by urease [28]. The data from this cropping cycle, in which there was no application of nitrogen rates, showed that the intercropping of soybean and Aruana Guinea grass was better than soybean–Congo grass in maintaining microbial N. This may result from the better adaptations of the roots of soybean and Aruana Guinea grass because during plant growth, the roots of the two crops become interwoven with each other, and the competition for root nutrition is relatively intense in cereals. The competitiveness of cereal crops in obtaining soil nitrogen is further strengthened, and the obstacle of "nitrogen repression" of legume root nodules is reduced, so both soybean and Aruana Guinea grass show obvious MBN advantages [25].

After the application of nitrogen, in the first maize crop, the intercropping systems affected the MBN content in different ways (Table 3). While the maize–Aruana Guinea grass intercropping increased microbial N (+165%) with the application of 50 kg ha$^{-1}$ of nitrogen, the maize–Congo grass intercropping impaired the MBN content due to nitrogen fertilization. It is noteworthy that, as observed in the MBC content, the maize–Aruana Guinea grass intercropping proved to be efficient at lower N rates, but rates greater than 100 kg ha$^{-1}$ nullified the benefits of this intercropping system to the MBN content. Such an increased MBN content was reflected in an increase in the total Nmic:Ntotal ratio. Thus, maize–Aruana Guinea grass intercropping could improve the quality of nitrogen added to the soil with little nitrogen supply. Moreover, an increase in N rates also impaired the enzymatic activity in intercropping in this season. Thus, as observed by [7], in some cases, the nitrogen availability interferes with the efficiency of nitrogen fixation, and it seems that the less N available for the plants, the more efficient this process becomes.

Fertilization with 100 and 150 kg ha$^{-1}$ of nitrogen reduced the urease activity in the intercropping systems maize–Aruana Guinea grass and maize–Congo grass, similarly to the MBN content. This may have been because the higher nitrogen level available in the intercropping systems was sufficient to inhibit UA [6,7,24]. Compared to the other cropping systems, nitrogen fertilization in maize–Aruana Guinea grass intercropping impaired UA (Figure 4e,f), with an expressive reduction in enzymatic activity at higher nitrogen rates (Table 3). The increase in nitrogen availability in the soil caused by the increase in nitrogen supply may have reduced the need for nitrogen mineralization and, thus, there was less UA [6].

Soybean–Congo grass intercropping in the second soybean crop showed an increase in soil MBN with increasing nitrogen rates, indicating the synergy of plants intercropped with adequate nitrogen rates (Table 2 and Figure 4a). Such an increase in MBN may have favored UA increase under the same conditions (Figure 4d), since, with an increase in nitrogen incorporated into the microbial biomass, there is greater availability of substrate for the enzyme to act [2,23,24,28].

## 5. Conclusions

Our results showed two important implications for grain production systems exploiting the synergy between suitable nitrogen fertilization rates and intercropping systems with tropical grasses. First, excess nitrogen fertilization can impair microbial carbon and nitrogen accumulation in the soil to the point of nullifying the benefits of intercropping systems with soybeans or maize and tropical grasses. Conversely, adding proper N rates to each intercropping system enhances the positive effects of tropical grasses on microbial carbon and nitrogen accumulations, reducing stress on the microbial community and increasing nutrient incorporation efficiency into microbial biomass. Therefore, the microbial community response to carbon and nitrogen cycling must be considered to adjust nitrogen fertilization rates for each cropping system, thus promoting homeostasis of the soil microbial community through synergy between tropical grasses and nitrogen fertilization.

**Author Contributions:** Conceptualization, K.B. and L.A.F.V.; methodology, K.B. and L.A.F.V.; formal analysis, K.B. and L.A.F.V.; investigation, K.B. and L.A.F.V.; resources, K.B.; data curation, K.B. and L.A.F.V.; writing—original draft preparation, L.A.F.V.; writing—review and editing, K.B. and L.A.F.V.; visualization, K.B. and L.A.F.V.; supervision, K.B. and L.A.F.V.; project administration, K.B.; funding acquisition, K.B. All authors have read and agreed to the published version of the manuscript.

**Funding:** This research was funded by the São Paulo Research Foundation—FAPESP (process 2017/50339-5 and process 2019/02387-6).

**Data Availability Statement:** The data that support this study are available in the article.

**Acknowledgments:** The authors are grateful to FAPESP (São Paulo State Research Support Foundation) for providing financial support.

**Conflicts of Interest:** The authors declare no conflict of interest.

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
