# Peer review of "Tropical Grasses—Annual Crop Intercropping and Adequate Nitrogen Supply Increases Soil Microbial Carbon and Nitrogen"

_agronomy, doi:10.3390/agronomy13051275_

Round 1

Reviewer 1 Report

Specific comments

As a researcher in the field of farming, I am very interested in your work. I have looked thoroughly at your article and I see that you did a lot of work on it.

However, There are some problems in the article that need to be solved, if I understand your description correctly. As far as I see, the paper can be accepted if the points below are dealt with appropriately.

Abstract

1. abstract focuses on a description of the study background and the study design, with less description of the conclusions and lack of specific data on the results of the study.

2. Line 12, “has” should read “have”.

3. Lines 14-15, delete and suggest to include it in materials and methods.

4. Line 25, “in short time” should insert “a”.

5. Lines 21-26. The results are not supported by specific data and the conclusions are ambiguous.

6. Lines 21-26,please state precisely whether the nitrogen supply should be adequate or appropriate.

Introduction

7. Line 31, What is the relationship between rising incomes and food demand at the worldwide level.

8. Line 35, “be also” should change “also be”.

9. Line 31-40, revise the introduction to better align with the content of the article.

10. Line 42, “world” should insert “the”, became “the world”.

11. Line 47. References [5, 6, 7, 8], perhaps you can use [5-8] instead.

12. Line 48, where the effect of crop intercropping with pasture on soil microbial activity and soil properties is not described.

13. Line 53,the location of the reference is unreasonable.

14. Line 54, soybean as a legume crop with nitrogen fixation, needs to be added regarding the effect of fertilizer application on the nitrogen fixation capacity of intercropped soybean.

15. There is no need to introduce too much other content, more focused on the research progress of intercropping system and nitrogen application on microorganisms.

16. Line 63,[5,11)should be changed [5,11].

17. Suggest expressing the purpose and significance of the research at the end of the introduction.

Materials and Methods

18. Line 78, should delete “in a”.

19. Line 89. Please check the format of writing the names of references.

20. Line 94: No spaces in the middle of 28 %.

21. Line 94,“dm-3” should be written as “dm-3”.

22. Lines 88-96. The basic characteristics of soils are recommended to be presented in a table format for easy reading.

23. Lines 88-101, at the beginning of the experiment, tillage, raking, and ash application were carried out. Is the soil related data measured before or after these operations? It is recommended to provide detailed explanations or adjust the order of expression.

24. The word legend for the lowest temperature in Figure 1 is incomplete.

25. Line 106, the experimental design does not clearly indicate whether soybean or maize is intercropped with both grasses in a year or one grass in a year.

26. Line 109, does not explicitly describe the varieties of soybeans and maize.

27. Lines 127 and 137, "0.45m" should be changed to "0.45 m".

28. Line 135, "0.90m" should be changed to "0.90 m".

29. Figure 1 is missing error bars.

30. N0 in Figure 2 considers whether the format is wrong or not, and whether a space is needed in the middle.

31. Line 141: The format of KD170 is wrong, and spaces are added between numbers and letters.

32. Line 147, “4oC” should be written as “4oC”.

33. It is recommended to indicate the data in Figure 2 for easy reading.

34. In the second image of Figure 2, it is recommended to use the same color with the same nitrogen application rate, which will be more neat.

Results

35. Line 187, there was no nitrogen application when soybean and grass were intercropped, and only nitrogen was applied when corn and grass were intercropped.  However, in the result analysis, there was a change in nitrogen application when soybean and grass were intercropped, which was inconsistent with the experimental design.

36. Lines 188-189,Maize-Aruana Guinea grass intercropping in the absence of nitrogen supply showed the highest MBC content in the first maize crop (Table 3)This statement is inconsisitent with the data in table 3 .It is advised to check and modify.

37. In Table 1, (1) is no description in the table note, and the specific meaning of the representative is not known.

38. The data in Tables 3 and 4 are consistent and fertilizer application does not produce any change in soil properties, which is not material to production practice, what was the purpose of doing the study?

39. Line 221, where 56 kg ha-1 of nitrogen is mentioned, whereas in the experimental design 50 kg ha-1 was applied. inconsistent.

40. Lines 282-284,Maize-Aruana Guinea grass intercropping at the nitrogen rate of 50 kg ha-1 showed higher qCO2 (2.3 times) in the first maize crop (Table 3). This sentence is also not reflected in Table 3.

41. Lines 308-309,In the first maize crop (Table 3), MBN increased linearly in maize monoculture as a function of the nitrogen rate increases as side-dressing (Figure 4b). According to the data in Table 3,this sentence should be changed toIn the first maize crop (Table 3), MBN decreased linearly in maize monoculture as a function of the nitrogen rate reduces as side-dressing (Figure 4b).

42. Line 328, chart (b) origin of curve has a little problem.

43. Lines 352-354,In the first maize crop, in the absence of nitrogen supply, UA in maize-Aruana Guinea grass intercropping was statistically equal to that in maize monoculture (Table 3). It can be seen from the data in Table 3 that UA under the two planting modes is not equal,and there are differences.

44. Lines 377 and 402, "100 and 150 kg ha-1" and "100 kg ha-1 and 150 kg ha-1" are recommended to be consistent.

45. In the results and analysis, we should focus on the differences between the data.

46. Why is the format of Table 1 different from that of Table 2,3,4?

47. In Table 2,3,4, please add the abbreviated form of Urease activity.

48. In Figure 3, What is the factor that determines whether you choose the first crop or the second crop?

49. The straight line in Figure 4 has a wrong trend and should be replaced with a downward line.

50. The font in Figure 3,4,5 is wrong and should be changed to The Times New Roman.

51. In Table 1, it is recommended to keep the same number of digits in the data. And the qMic font format is different from other content.

52.  In Table 1, it is recommended to explain (1).

53. In Figure 3, some units have different font sizes for small footmarks.

54. The article only mentions that corn is treated with different gradients of nitrogen fertilizer. Why does Table 2 show that soybeans are also treated with different gradients of nitrogen fertilizer?Why only apply different gradients of nitrogen to the second crop of soybeans?

Discussion

55. Line 402, please add relevant literature to support the findings.

56. Line 427, there are formatting issues in the article, such as ‘application, (Table 1)’.

57. Line 453,‘higher N demand for these intercropping systems. [26] reported that Congo grass showed, there is an error in this line.

58. Line 453, not literature is reported, but the result of research or survey by related people, please add related authors.

59. The analysis of the results still dominates the discussion, and the discussion of the results is lacking, and more literature needs to be added to validate the results.

60. Lines 476 and 479,The NBM content should be changed The MBN content .

61. Line 497, “because lower” should insert “of”.

62. Lines 497-498,This may have been because lower nitrogen available under intercropping system was sufficient to inhibit UA High nitrogen levels can be added after this sentence to further inhibit UA.This expression is more consistent with the study of this paper.

63. Line 502, suggest paying attention to the details of the article.

64. Line 509, please cite the references in order of size.

65. Line 510, this paragraph is more like the relevant conclusion of this study.

66. When adding references, they should be added in the order of references, so they should be changed to[4,24,25,27].

67. There are too few references in the discussion, please add them.

68. The discussion section may be divided for reading.

Conclusion

69. Adequate supply of nitrogen fertilizer ? Which gradient is specific in nitrogen treatment ? The conclusion is ambiguous.

70. Line 528,and consequently reduced the NBM contentshould be modified and consequently reduced the MBN content.

71. The conclusion part summarizes the different effects of nitrogen sufficiency and deficiency in intercropping, but lacks specific data to support it.

72. Line 532, “in short term” should insert “the”.

73. The innovation of this experiment should be explained simply and clearly.

74. It is suggested to add some advantages and disadvantages compared with other studies.

75. It is suggested to supplement some limitations or deficiencies, as well as prospects for future research.

References

76. Article 12 literature why there is no title.

77. Please note that the format of references is consistent with journal requirements, and check the writing of the person’s name, journal name, and punctuation.

78. Line 581, Line 583, Line 587, Line 589, 599, 600, reference years need to be capitalized.

79. According to the periodical requirements, do you need to add DOI?

Author Response

Dear MDPI Agronomy Editorial Office

We are sharing the revised version of the manuscript ID agronomy-2346768 entitled "Tropical grasses - annual crop intercropping and adequate nitrogen supply increases soil microbial carbon and nitrogen". The comments and suggestions of the reviewers were very constructive and the manuscript has benefited from these concerns. We tried to make our best to meet the reviewers. We feel that the revised manuscript has significantly improved. Please find attached the response to comments, doubts and suggestions to Reviewer 1.

Reviewer 2 Report

Recommended edits to abstract: In Brazil, grain crops under no-till soybean-maize succession has reduced biodiversity and carbon input into soil. Intercropping is a promising approach to address these problems. This study aimed to evaluate the microbiological quality of soil under conventional and intercropping soybean-maize production, this succession as a function of tropical grass and nitrogen fertilizer uses. The treatments were arranged in a randomized complete block design and a split-plot scheme, with four replications. The main plots consisted of cropping systems: soybean monoculture – maize monoculture; soybean intercropped with Aruana Guinea grass (Megathyrsus maximus cv. Aruana) – maize intercropped with Aruana Guinea grass; and soybean intercropped with Congo grass (Urochloa ruziziensis cv. Comun) – maize intercropped with Congo grass. The subplots consisted of nitrogen rates (0, 50, 100 and 150 kg ha-1) applied as side dressing in rows of maize and tropical grass in the autumn-winter season. Maize or soybean intercropping with tropical grasses supplied with suitable nitrogen rates favoured the sequestration entry of microbial carbon and nitrogen, stimulated enzymatic activity, and reduced C-CO2 loss. However, the benefits from nitrogen rates and intercropping systems on increasing microbial carbon and nitrogen varied between soybeans and maize crops. We concluded, that the intercropping systems evaluated here can improve soil microbiological quality rapidly in short time, providing as long as the nitrogen supply is adequate.

Major concern: Why are there no error bars on any of the graphs? There are – as I read it – four replicates. All we are given is a log-transformed CV. This really hides the variability and is unacceptable to this reviewer. We need a more transparent expression of the variability observed in these experiments.

Minor edits/comments:

Line 34 – replace “more than 784” with “nearly 800” (we do not know the population to three significant digits – the 8% number on which this is based is 1 significant digit, so 1 sig digit in the computed result is far more reasonable.

This reviewer recognizes the effort in that first paragraph is to contextualize, but it is not strong. The paper might be improved by simply starting with Line 41 (simply delete lines 31 – 40).

Line 45 – 46: Two issues. (1) In what sense? (2) Define “intercropping systems” upon first use. Suggested wording that may address both issues:

Replace “In this sense, intercropping systems may…” with “One way of addressing these concerns while maintaining the row-crop output of these systems is through the establishment and maintenance of another plant between the rows of corn or soybean, a practice known as intercropping. These intercropping systems may…”

Paper would be strengthened by addressing yield impacts of treatments. Not a requirement for publication, but would have been good.

Very good English overall - see earlier comments for a few suggestions to improve.

Author Response

Dear MDPI Agronomy Editorial Office

We are sharing the revised version of the manuscript ID agronomy-2346768 entitled "Tropical grasses - annual crop intercropping and adequate nitrogen supply increases soil microbial carbon and nitrogen". The comments and suggestions of the reviewers were very constructive and the manuscript has benefited from these concerns. We tried to make our best to meet the reviewers. We feel that the revised manuscript has significantly improved. Please find attached the response to comments, doubts and suggestions to Reviewer 2.

Best regards!

Round 2

Reviewer 1 Report

I am quite satisfied with the author's revision

Reviewer 2 Report

Response to review is acceptable.